# Tumor Immune Microenvironment and Its Clinicopathological and Prognostic Associations in Canine Splenic Hemangiosarcoma

**DOI:** 10.3390/ani14081224

**Published:** 2024-04-18

**Authors:** Chiara Brachelente, Filippo Torrigiani, Ilaria Porcellato, Michele Drigo, Martina Brescia, Elisabetta Treggiari, Silvia Ferro, Valentina Zappulli, Monica Sforna

**Affiliations:** 1Department of Veterinary Medicine, University of Perugia, Via San Costanzo 4, 06126 Perugia, Italy; chiara.brachelente@unipg.it (C.B.); marty.brescia@gmail.com (M.B.); monica.sforna@unipg.it (M.S.); 2Department of Comparative Biomedicine and Food Science, University of Padua, AGRIPOLIS, Viale dell’Università 16, 35020 Legnaro, Italy; filippotorrigiani6@gmail.com (F.T.); silvia.ferro@unipd.it (S.F.); valentina.zappulli@unipd.it (V.Z.); 3Department of Animal Medicine, Production and Health, University of Padua, AGRIPOLIS, Viale dell’Università 16, 35020 Legnaro, Italy; michele.drigo@unipd.it; 4Clinica Veterinaria Croce Blu, via San Giovanni Bosco 27/C, 15121 Alessandria, Italy; e.treggiari@gmail.com

**Keywords:** tumor microenvironment, tumor-infiltrating lymphocytes (TILs), canine splenic hemangiosarcoma (HSA), immunohistochemistry, prognostic factors

## Abstract

**Simple Summary:**

This retrospective study explores how tumor cells impact the surrounding tissue, creating what is known as the tumor microenvironment. This environment, consisting of various cells and structures, can influence cancer progression. Researchers focus on understanding the role of immune cells, specifically tumor-infiltrating lymphocytes (TILs), in this environment. In this investigation, 56 cases of canine splenic hemangiosarcoma (HSA) were studied to analyze TILs’ impact on the tumor’s histologic characteristics and how it affects survival. The findings highlight the association between certain immune cell distributions and factors like tumor-related death, survival, and metastasis. Overall, this study delves into the immune environment of canine splenic hemangiosarcoma, offering insights into potential prognostic factors.

**Abstract:**

Tumor cells can induce important cellular and molecular modifications in the tissue or host where they grow. The idea that the host and tumor interact with each other has led to the concept of a tumor microenvironment, composed of immune cells, stromal cells, blood vessels, and extracellular matrix, representing a unique environment participating and, in some cases, promoting cancer progression. The study of the tumor immune microenvironment, particularly focusing on the role of tumor-infiltrating lymphocytes (TILs), is highly relevant in oncology due to the prognostic and therapeutic significance of TILs in various tumors and their identification as targets for therapeutic intervention. Canine splenic hemangiosarcoma (HSA) is a common tumor; however, its immune microenvironment remains poorly understood. This retrospective study aimed to characterize the histological and immunohistochemical features of 56 cases of canine splenic HSA, focusing particularly on tumor-infiltrating lymphocytes (TILs). We assessed the correlations between the lymphocytic response, the macroscopic and histological characteristics of the tumor, and the survival data. Our study demonstrated that FoxP3 distribution was associated with tumor-related death and survival, while the CD20 count was associated with metastasis. This study provides an in-depth characterization of the tumor immune microenvironment in canine splenic HSA and describes potential prognostic factors.

## 1. Introduction

Hemangiosarcoma (HSA) is a malignant tumor of dogs presenting in various anatomical forms with distinct biologic behaviors, based on their cell of origin. Recent studies suggest that cutaneous hemangiosarcoma likely originates from transformed mature endothelial origin, exhibiting a comparatively better biologic behavior with rare metastases, whereas other forms of HSA likely arise from the precursor (pluripotent) endothelial cell in the bone marrow, subsequently migrating and colonizing organs with viable microenvironments (spleen, liver, right atrium) [1,2,3]. In the spleen, HSA is the most common neoplasm, representing 50–74% of all splenic malignancies in dogs [4,5]. It is characterized by aggressive behavior and early metastases mainly affecting the liver, omentum, mesentery, lungs, and brain. The survival time usually ranges between 19 days and 260 days [6]. Recent comparative genomic studies have revealed a shared genomic landscape between canine HSA and human angiosarcoma of the breast and viscera, suggesting that the dog can be considered a useful model for the study of human angiosarcoma [3,7,8,9,10,11,12,13,14].

Research studies which focused on identifying histologic prognostic factors and potentially establishing a grading system for canine splenic HSA were conducted in two studies by Moore [15] and Ogilvie [16]. These investigations considered parameters such as tumor differentiation, nuclear pleomorphism, tumor necrosis, and mitotic count. However, the association with prognosis (overall survival, OS) of affected dogs was reported in the study by Moore et al. on univariate analysis only and, as a result, histological grading of canine HSA is not routinely performed [17,18]. A staging system is available as well, similar to the grading system, whereby dogs classified at stage I tend to exhibit longer survival times compared with those at stages II and III [16,19,20]. However, the practical use of clinical staging has been a topic of debate in the scientific literature [18,21,22,23]. Consequently, additional prognostic features are needed. The investigation of the tumor microenvironment (TME), and in particular the tumor immune microenvironment (TIME), has provided in recent years many useful insights. These insights not only enhance our understanding of tumor biology but also identify potential mechanisms underlying immune response dysregulation and therapeutic intervention strategies [24,25,26]. The immune response, in addition to protecting the individual against foreign invaders, is fundamental in contrasting mutated endogenous cells such as cancer cells. Proof of this is the ability of the immune system to induce spontaneous regression of some tumor types such as papillomas and histiocytomas [27,28].

Tumor-infiltrating lymphocytes (TILs) are immune cells, predominantly T-cells and B-cells, whose presence, density, and functional status have been associated with prognosis and treatment response in various cancers [29,30,31,32]. Indeed, tumors become infiltrated with different cell populations of both the innate (neutrophils, macrophages, natural killer cells, mast cells, dendritic cells) and adaptative (B- and T-cells) immune response and, altogether, immune cells can either suppress tumor growth or promote it [33]. CD8+ T cells are the most effective in killing neoplastic cells in an antigen-specific manner, by activating apoptosis or secreting cytotoxic molecules (e.g., granzyme, perforin). Cytotoxic T-cells must, however, be activated by dendritic cells in lymph nodes or in organized tertiary lymphoid structures within or adjacent to the tumor tissue [34]. Other cells, such as NK cells, can help kill neoplastic cells without the priming of dendritic cells or antigen-presenting cells [35]. Also, CD4+ Th1 cells promote anti-tumor responses by secreting cytokines important for CD8+ T cell proliferation, macrophage recruitment, and activation. On the other hand, M2 macrophages and myeloid-derived suppressor cells (MDSCs) can promote tumor progression through the inhibition of the immune response and the induction of stromal cell proliferation, vascularization, extracellular matrix (ECM) deposition, and cell migration [36]. Additionally, CD4+ T cells that express the forkhead box P3 (FoxP3) transcription factor function as regulatory T (Treg) cells and either directly or indirectly induce immunosuppression, both promoting immune tolerance and actively inhibiting anti-tumor immune responses. The suppression of the immune response occurs through various mechanisms such as the secretion of immunosuppressive molecules (e.g., IL-10 and TGF-β) and modulation of the APC function (e.g., via CTLA-4–CD80/86 interactions) [37]. In addition, FoxP3 cells can promote the recruitment of immunosuppressive cells such as myeloid-derived suppressor cells (MDSCs) and M2 macrophages [38]. FoxP3 expression was found to be significantly associated with relapse in human cutaneous angiosarcoma [39]. Strengthening the comparative role of veterinary oncology, these cells have been found to be present in many canine tumors and in particular in the TME of HSA in dogs, where they are associated with a poorer prognosis, suggesting that they can also contribute to the immune evasion mechanisms adopted by tumor cells [40].

B cells are involved in humoral responses, but recent studies have discovered their role in modulating anti-tumor immune responses, with a dual and sometime opposite role, depending on the context of tumor microenvironment. In the context of anti-tumor immunity, CD20 lymphocytes facilitate antibody-dependent cellular cytotoxicity (ADCC) and complement activation, as well as act as antigen-presenting cells (APCs), presenting tumor antigens to cytotoxic T lymphocytes [41,42].However, it has recently been demonstrated that B cells dampen the immune response through the production of immunosuppressive cytokines (IL-10, IL-35, TGF-β) in a way similar to Tregs [43].

The responses activated by the immune system submit tumor cells to significant selective pressure. Indeed, cancer cells have developed strategies to evade the immune response, such as, for example, the downregulation of the expression of major histocompatibility complex (MHC) molecules [44,45]. Other strategies adopted by tumor cells are the upregulation of the so-called immune checkpoint molecules, such as cytotoxic T-lymphocyte-associated protein 4 (CTLA-4) and programmed cell death protein 1 (PD-1) with its ligand (PD-L1) [44]. These molecules cause an inactivation of T cells, preventing them from recognizing and killing tumor cells. The crucial role of these molecules has been supported by the successful application of cancer immunotherapy using drugs (immune checkpoint inhibitors), reverting the inhibition of T cells and restoring the ability to recognize and eliminate cancer cells [44]. Also, it appears that the responsiveness to these therapies depends on many still unknown factors, including some intrinsic features of the TME [46,47]. Therefore, the identification of biomarkers able to define the prognosis and generally the clinical outcome, as well as to predict the response to immunotherapies, is the subject of extensive studies and clinical trials. In this context, an association between the number and type of TILs and a favorable clinical response has been demonstrated in human patients with melanoma and colorectal cancer, where higher numbers of CD8+ TILs in pre-treatment biopsies of these patients are associated with better responses to anti-PD-1 or anti-PD-L1 therapies [48,49,50,51,52,53]. Vice versa, this has not been demonstrated in renal cell carcinoma [54,55]. These differences emerging in different tumor types emphasize that the composition of the TIME does not regulate a dichotomous either totally pro-tumoral or anti-tumoral environment and that TIME is not a static milieu but depends on the ever-evolving interactions between immune cells, tumor cells, and stromal cells.

Studies attempting the characterization of TME/TIME and clinicopathological and prognostic associations in human angiosarcomas have been recently published [56,57,58] considering that angiosarcomas are recognized as tumors with a high mutational burden, therefore able to produce altered proteins, which might be recognized as foreign antigens and elicit an immune response [59]. The use of immune checkpoint inhibitors, namely anti-CTLA-4 and PD-L1 blocking agents, was reported with promising results in human angiosarcoma by Florou and others [60,61,62]. However, the majority of the studies were performed on a small number of cases and on cutaneous, predominantly UV-associated angiosarcoma, whereas the other types of angiosarcomas, including the visceral ones, were less commonly investigated [63]. Because of the previously expressed similarities between canine and human HSA and the potential of the dog as a model for human beings, the incidence of canine splenic HSA provides an excellent opportunity to investigate the role of the immune response in visceral angiosarcomas. The few published studies are sporadic, yet an activation of the PD-1/PD-L1 axis has been reported in canine oral melanoma, osteosarcoma, HSA, mast cell tumor, mammary adenocarcinoma, and prostatic adenocarcinoma [64]. Gulay et al. have reported, in a murine syngeneic model for canine HSA, that hemangiosarcoma cells in vitro can induce M2 polarization of macrophages and upregulation of PD-L1 expression, supporting the role of immune cells in canine HAS as well [65].

This retrospective study aimed to characterize the TIME of canine splenic HSA. The association with clinicopathological data was calculated to determine the potential prognostic role of TILs and associated markers in this tumor.

## 2. Materials and Methods

### 2.1. Case Selection

Cases of canine splenic HSA were retrospectively selected from the pathology archives of the Department of Comparative Biomedicine and Food Science, University of Padova, and the Department of Veterinary Medicine, University of Perugia, from 2016 to 2019. Patient records were reviewed for breed, age at diagnosis, gender, tumor stage, overall survival, and date of last follow up or death. Follow-up information was collected through a phone interview with the referring veterinarians or through the collection of medical records data from internal cases. Primary tumor specimens from treatment-naïve dogs were used for hematoxylin and eosin (H&E) and immunohistochemical (IHC) staining. The inclusion criteria for the immunohistochemical evaluations were:histological diagnosis of HSA;in case of poorly differentiated tumors, a diagnosis of endothelial origin had to be confirmed by immunolabeling with vWF (FVIII) and/or CD31;available neoplastic tissue with an area >0.5 cm^2^.

Samples were excluded if they were characterized by fixation artifacts. In cases with more than one available histocassette, the most representative was chosen, avoiding samples with large necrotic areas or ulceration.

The overall survival was defined as the time from the first diagnosis/appearance to death for any cause. Tumor progression was defined as the appearance of new abdominal lesion(s) or metastases after splenectomy. Since the number of new abdominal lesions and metastasis was not numerous, we considered them altogether (“metastasis”) for statistical analysis. The clinical outcome of dogs that died because of the tumor was considered ‘unfavorable’, while the one of patients alive or dead for causes unrelated to HSA was considered ‘favorable’.

### 2.2. Immunohistochemistry

From formalin-fixed and paraffin-embedded (FFPE) samples, 5-μm sections were cut and mounted on poly-L-lysine-coated slides, which were then dewaxed and dehydrated. Immunohistochemistry was manually performed on serial sections with antibodies raised against Iba-1, FoxP3, CTLA-4, CD3, and CD20, as previously reported [66,67,68]. The protocol details and the origin of antibodies are summarized in Table 1.

Iba-1, FoxP3, CTLA-4, CD3, and CD20 were used in previous studies from our group investigating the TIME of canine melanocytic tumors [68,69,70]. In particular, the FoxP3 FJK-16s clone is reported to cross-react with the canine antigen [71], whereas anti-CTLA-4 antibodies were stated to cross-react with the canine tissue and to be suitable for FFPE material in the datasheet provided by the manufacturer. Positive controls were obtained from canine reactive lymph nodes for all the five antibodies used in this study; negative controls were run, omitting the primary antibody and incubating control sections with TBS. The expected reactivity was membranous/cytoplasmic for CD3 and CD20, cytoplasmic for Iba-1 and CTLA-4, and nuclear for FoxP3.

For each antibody, positive cells were manually counted in five consecutive high power fields (HPF, 40× objective, 10× ocular, FN = 22, total area per field = 0.237 mm^2^) [72] starting from hot spots, and then the mean value was calculated; positive cells were considered only if they were intratumoral. Positive cells in the peripheral areas of the tumor or the immediate peritumoral areas were not quantified, but their presence was recorded.

The following parameters were evaluated for the immunohistochemical markers:

Iba-1, CTLA-4, FoxP3, CD3, CD20: “Distribution” (Model of distribution): 0 = absent, 1 = focal, 2 = multifocal, 3 = diffuse.CD3, CD20: “Groups” (Arrangement of positive cells): 0 = single cells, 1 = small aggregates, 2 = massive aggregates (≥30 positive cells). If more than one type of arrangement of inflammatory cells was present in the same case, the final score was assigned to the more commonly represented type.Iba-1, CTLA-4, FoxP3, CD3, CD20: “Quantity”: average number of positive cells per unit area (0.237 mm^2^). If there were massive lymphocyte aggregations, the wording “>100” was used.

All immunohistochemical slides were separately reviewed by two experienced pathologists (IP, MS), who were blinded to the clinical data of patients. Any disagreement was resolved through re-review and discussion until a final agreement was reached.

### 2.3. Statistical Analysis

The immunohistochemical expression of the tested markers was analyzed according to clinical data such as the clinical stage, the presence of metastasis, the overall survival time, and the outcome (tumor-related death).

Methodology–continuous data were assessed for normality of distribution with the Shapiro–Wilk test. Descriptive statistics and graphical distribution for the immunohistochemical expression of the tested markers are provided to describe their distribution pattern with respect to clinical stage (i.e., I, II, and III) and the presence of metastasis (i.e., yes/no); the correlation between the quantitative immunohistochemical expression of the tested markers was investigated by means of the Spearman rank-order correlation coefficient.

The association between the quantitative immunohistochemical expression of the tested markers and clinical stage was analyzed using the Kruskal–Wallis test by ranks followed by post hoc analysis by means of the pairwise median test. The Wilcoxon Mann–Whitney non-parametric test was used to analyzed the difference of the immunohistochemical expression of the tested markers with respect to the metastatic disease.

Finally, a survival analysis was conducted to assess the overall survival rate for dogs with follow-up data recorded according to the immunohistochemical distribution score of the test markers (i.e., absent, focal, multifocal, diffuse) by means of the construction of Kaplan–Meier curves and the log-rank test for their global and pairwise statistical comparison.

For all statistical analyses, values of *p* < 0.05 were considered significant.

## 3. Results

### 3.1. Patients Characteristics and Immunohistochemical Findings

A total of 56 cases of splenic HSA were included in the analysis. The median age of the patients was 10.7 years (interval min–max: 4–16 years). Most patients were male (34/56, 60.7%; 31 males and 3 neutered males), resulting in a male-to-female ratio of 1:6. While the majority of affected dogs were mixed breed, overrepresented breeds included Labrador retrievers and German shepherds. Appendix A show signalment, follow-up information, and immunohistochemical findings of the 56 cases of canine splenic HSA examined in this study, respectively.

Immune cells were present in the majority of samples in varying proportion and distribution (Figure 1).

In particular, CD3+ lymphocytes were present in all tumors examined, with an average quantity of 35.5 cells/0.237 mm^2^. Regarding the aggregation pattern, in 31 tumors (56.4%), CD3+ cells were mostly distributed as single cells, in 6 tumors (10.9%) they were arranged in small aggregates, and in 18 tumors (32.7%) in large aggregates.

CD20+ lymphocytes were present in 44/56 (21.4%) tumors, with an average quantity of 19.3 cells/0.237 mm^2^. Regarding the aggregation pattern, in 33 tumors (58.9%), CD20+ cells were mostly distributed as single cells, in 11 tumors (19.6%) they were arranged in small aggregates, and in 12 tumors (21.4%) in large aggregates.

Iba1+ cells were present in 50/56 (89.3%) tumors, with an average quantity of 23.3/0.237 mm^2^.

FoxP3+ cells were present in 52/56 (92.9%) tumors, with an average quantity of 3.7/0.237 mm^2^.

CTLA-4+ cells were present in 36/56 (64.3%) tumors, with an average quantity of 1.1/0.237 mm^2^.

The Spearman correlation coefficient showed a weak positive correlation between the quantity of FoxP3+ cells and CTLA-4+ cells (Rho = 0.395; *p* = 0.003). Similarly, a weak positive correlation was found between the quantity of FoxP3+ cells and CD3+ cells (Rho = 0.394; *p* = 0.003) as well as the quantity of CD20+ cells and CD3+ cells (Rho = 0.352; *p* = 0.008)

No correlation between the number, distribution and groups of immune cells and the patient characteristics (breed, age, gender) was noted.

Representative images of immunohistochemical labelling are shown in Figure 2 and Figure 3.

### 3.2. Prognostic Significance of Immunohistochemical TIME-Related Markers in Canine Splenic Hemangiosarcoma

All 56 dogs had follow-up data available. Seven dogs were diagnosed as stage I (7/56, 12.5%), 21/56 (37.5%) as stage II, and 12/56 (21.4%) as stage III. For 16 patients (28.6%), this information could not be retrieved from the medical records. Twenty-seven dogs (48.2%) developed metastatic disease, whereas ten dogs (17.8%) had no detectable metastases. For 19 animals (33.9%) this information was not available. At the end of the study, 38 dogs (67.5%) were dead because of tumor-related causes with a median OS of 221.6 days. Survival analysis showed no significant difference in OS in terms of patient signalment (breed, age, gender). The distribution of CD3+ cells, CD20+ cells, Iba-1+ cells, FoxP3+ cells, and CTLA-4+ cells was not correlated with the clinical stage, neither was the aggregation pattern of CD3+ or CD20+ cells. However, although not statistically significant, Iba-1 and CTLA-4 quantity increased with the clinical stage (Figure 4).

In terms of survival, patients with a focal (*p* = 0.040) and multifocal (*p* = 0.033) distribution of FoxP3+ cells exhibit significantly decreased overall survival and higher risk of tumor-related death, compared with those with absent FoxP3+ cells (Figure 5).

CTLA-4 distribution showed a trend of association with a shorter survival (*p* = 0.066) although the difference was not statistically significant.

Furthermore, CD20+ cells showed statistically higher median values in association with metastatic disease (*p* = 0.013) (Figure 6).

Dogs with metastatic disease showed generally higher median values of Iba-1, FoxP3, CTLA-4, and CD3+ cells; however, no statistically significant differences were noted (Figure 7).

## 4. Discussion

Hemangiosarcoma is the most common and most aggressive primary tumor of the spleen in dogs, and clinical signs include the acute onset of hemorrhage due to associated organ rupture and hemoabdomen; this is usually related to a high metastatic rate to other internal organs, including peritoneum, liver, and lungs. Metastases are typically hematogenous or via transabdominal implantation. Human angiosarcoma has been suggested as an immunogenic tumor [73] with tumor cell expression of PD-1 and PD-L1 associated with high-grade, poorly differentiated tumors and a worse prognosis. However, the role of the immune cells and the expression of immune checkpoint molecules as prognostic biomarkers has not been well established in canine splenic HSA. The present study aimed to explore the clinicopathological and immunohistochemical characteristics of TIME of splenic HSA in dogs. The median age at diagnosis was 10.7 years, with a male preponderance and an overrepresentation of Labrador retriever and German shepherd dogs as affected breeds, in line with what has already been reported in previous studies [74,75,76,77,78,79].

Our results showed the presence of an immune cell infiltration in canine splenic hemangiosarcoma, as demonstrated by the prevalence of the different immune cell populations in our case series. It could be hypothesized, therefore, that canine splenic HSA can be considered an immunogenic tumor.

We could not identify a prognostic significance of the majority of the variables investigated. This could be due to a small sample size, likely too little to detect a significant association. Nevertheless, the correlation found between several immune cell populations, and in particular between FoxP3+ and CTLA-4+ cells, suggests a potential role of immune cells in canine splenic HSA where mechanisms of tumor evasion are activated. This seems to be supported by the association of the multifocal distribution of FoxP3+ cells with tumor-related death and shorter survival, suggesting a prognostic association of Tregs. The modulation of tumor immune microenvironment exerted by Tregs could play a role in oncological progression of canine splenic hemangiosarcoma. The accumulation of Tregs in the tumor could exert suppressive effects on immune responses, by promoting an immunosuppressive milieu, conducive to tumor immune evasion, progression, and metastatic spread.

FoxP3 expression in human cutaneous angiosarcoma was reported to be associated with disease relapse by Gambichler et al. [39], and a higher number of FoxP3+ cells was found in the peripheral blood of patients with HSA [56], demonstrating that the dysregulation of Tregs plays an important role in the progression of angiosarcoma. In dogs, FoxP3 expression was described in several tumors, such as gliomas [80] and histiocytic sarcomas [81]. Additionally, it was found to be an independent predictor of death in canine melanoma [68] and associated with poor prognostic factors, such as high histological grade, lymphatic invasion, and necrosis in canine mammary carcinomas [82]. Dogs with osteosarcomas were shown to have significantly higher levels of circulating Treg and lower CD8/Treg ratio compared with healthy dogs with decreased CD8/Treg ratio, being associated with significantly shorter survival times [83]. It is therefore not surprising that in a tumor with a highly aggressive behavior such as splenic HSA, the activation of Tregs-mediated responses is associated with a worse outcome.

Immunological and genomic analysis has suggested clinically relevant differences among the different types of human angiosarcoma, revealing T-cell infiltrated microenvironments (CD3+ T-cells, CD8+ cytotoxic T-cells, CD4+ T-helper cells, and FoxP3+ T-regulatory cells) especially in “secondary” angiosarcomas, such as the ones caused by DNA damaging factors, namely ultraviolet light exposure, radiotherapy, or chronic lymphedema [61,84,85]. In view of an approach to increasingly personalized medicine, the characterization of the immune landscape of a tumor becomes an important factor in the identification and stratification of patients who are likely to benefit from immunotherapy-based treatment strategies. This is particularly important for HSA which, even in human medicine, has limited therapeutic possibilities and a poor prognosis and in which emerging data suggest that alternative approaches with the use of immune checkpoint inhibitors are effective [86]. Contradicting results emerged from a retrospective study comparing the clinical characteristics and the results of targeted exome sequencing, transcriptome sequencing, and immunohistochemistry analyses in human patients with histologically confirmed angiosarcoma treated with immune checkpoint blockade-base therapy, where it was demonstrated that neither PD-L1 expression nor presence of TILs at baseline appears necessary for a response to immune checkpoint blockade (ICB) therapy [86]. Indeed, despite the demonstration that distinct immune and mutational profiles are present across the spectrum of angiosarcomas [85] and that tumor mutation burden, PD-L1 overexpression, and an immunogenic tumor microenvironment have been suggested to predict the response to immunotherapy, no study demonstrated the efficacy of any single parameter alone, suggesting that an integrated approach using multiple targets in combination should be used [60,87,88,89,90]. Response to immune checkpoint inhibitors-based therapy seems to be higher in patients with UV-driven angiosarcoma, which is usually characterized by a high tumor mutational burden [91].

To the authors’ knowledge, no studies have been performed to demonstrate whether canine splenic HSA can be considered similar to human visceral angiosarcomas, which usually lack the high tumor mutational burden associated with ultraviolet (UV) damage mutational signature, typical of angiosarcoma of the head, neck, face, or scalp [84]. However, the demonstration, in our study, of the expression of CTLA-4 and its trend of association with a shorter overall survival, suggests the possible application of immune checkpoint inhibitors-based therapeutical strategies in association with the chemotherapy protocols used in such cases. This result should be confirmed in a clinical setting, in order to show a clinical response to immune checkpoint inhibitors in a selected cohort of patients affected by splenic HSA.

Our study showed that the quantity of CD20+ cells was significantly associated with the risk of metastatic disease. B lymphocytes participate in both humoral and cellular immunity, but their role in tumor responses remains controversial, with some studies showing that B cells can induce antitumor activity, while others have found that B cells may exert protumor functions due to their various immunosuppressive subtypes [92,93,94]. In human sarcomas, the composition of the immune cell populations and in particular B and T cells, have been demonstrated to better predict response to immune checkpoint inhibitors, compared with the classically accepted biomarkers, such as the tumor mutational burden and the PD-L1 expression [95]. A high density of B cells and the presence of tertiary lymphoid structures were found to be the strongest prognostic factors even in the context of high or low CD8+ T cells and cytotoxic contents in a study examining the gene expression profiles of 608 different subtypes of soft tissue sarcoma [95]. In a study comparing the TIME of primary (angiosarcomas developing in any anatomical sites with no known etiology) and secondary (angiosarcoma caused by DNA damaging factors such as UV exposure, radiotherapy, and chronic lymphedema) angiosarcomas, no differences in the median count of CD20+ B cells were noted [84]; however, visceral angiosarcoma showed a higher amount of CD20+ B cells compared with other tumors in the primary angiosarcoma group [84]. In a study investigating the correlation between the presence of CD20+ B cells and clinicopathological characteristics in human oral squamous cell carcinoma, it was found that peritumoral CD20^+^ B lymphocyte infiltration is associated with lymph node metastasis, suggesting that these cells can be considered as prognostic indicators [96]. Similar results were obtained in a case series of 97 canine melanocytic tumors, where a high infiltration of CD20^+^ TILs was associated with tumor-related death, the presence of metastasis/recurrence, shorter overall and disease-free survival, increased hazard of death, and development of recurrence/metastasis [70]. It is therefore not surprising that similar findings can be observed in canine splenic HSA.

Regarding the role of tumor-associated macrophages (TAMs), they have been shown to represent important cells in immune responses against cancer. The dichotomous model of macrophage polarization, obtained in vitro in mouse models, has long been used as a prototype of a Th1-type proinflammatory environment associated with the presence of M1 macrophages with a tumor-suppressing effect and a Th2-type anti-inflammatory environment associated with the presence of M2 macrophages with a tumor-promoting effect [97]. However, this M1/M2 dichotomy does not fully reflect the remarkable plasticity and complexity of macrophage polarization and additional phenotypes other than M1/M2 that have been discovered [98]. In dogs, TAMs were investigated in several tumors, such as canine and feline mammary tumors, glioma, osteosarcoma, lymphoma, and soft tissue sarcoma, and were attributed a negative prognostic significance [99,100,101,102,103,104]. CD204+ TAMs were found to be present in high numbers in canine visceral (splenic) HSA, with a predominantly M2-skewed phenotype, although no association could be demonstrated between histological parameters or clinical stage and TAM numbers or phenotype [65,105]. Tumor-associated macrophages would be important in tumor progression as well. Indeed, increased CD18+ monocytes numbers were found in HSA metastases compared with metastases from other tumor types, and this result, in association with an increased expression of the monocyte chemokine CCL2 by tumor cells, would support the hypothesis that overexpression of CCL2 and recruitment of large numbers of monocytes could explain the aggressive metastatic nature of canine HSA [106,107,108]. Our results partially confirm the published literature because, although no statistically significant differences in terms of prognosis were noted, the numbers of Iba-1+ cells increased with tumor stage. In the present study, Iba-1 was chosen as a pan-macrophage marker [109,110] as the investigation of the expression of M1/M2-polarization markers by TAMs was beyond the scope of this paper. Similarly, Kerboeuf et al. recently demonstrated a lack of association between histological parameters or clinical stage and TAM numbers or phenotype in canine HSA [105]. However, the authors of the study confirmed the presence of numerous macrophages and M2 macrophages (CD206+ cells) in tumor hot spots, particularly in splenic tumors, opening the way to the potential application of immunotherapies aimed at repolarizing TAMs in canine splenic HSA.

Our study, although being the result of a multi-institutional collaboration, has some limitations. Firstly, the evaluation of the distinct immune cell populations, although performed in a systematic and repetitive mode, as in other similar prognostic studies of this type, did not consider the relative density of the immune cells compared with the neoplastic cells, and this is not necessarily an established evaluation method. Secondly, this was a retrospective observational study that used medical records and telephone interviews and the sample size may not be enough to provide solid statistical results. Future multi-institutional (i.e., veterinary clinics and pathology labs) collaborations able to pool even more resources, patient data, and samples will be required to confirm the results of the present study and will be critical in building larger datasets capable of deepening our understanding of this complex disease.

## 5. Conclusions

In conclusion, our study confirms the presence of an immune response in canine splenic HSA and underscores the need of multi-institutional studies focused on the identification of prognostic factors in canine tumors. Survival outcomes of our case series were in line with previous studies, yet no prognostic significance of the majority of the immunohistochemical parameters investigated was noted. However, the multifocal distribution of Treg and higher counts of CD20+ cells had a negative prognostic impact with a higher percentage of tumor-related death and shorter survival and risk of developing metastasis, respectively.

## Figures and Tables

**Figure 1 animals-14-01224-f001:**
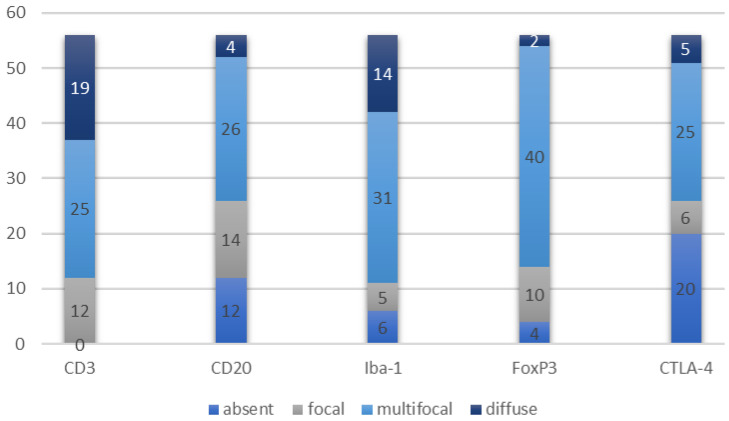
Prevalence of the populations of immune cells and their pattern of distribution.

**Figure 2 animals-14-01224-f002:**
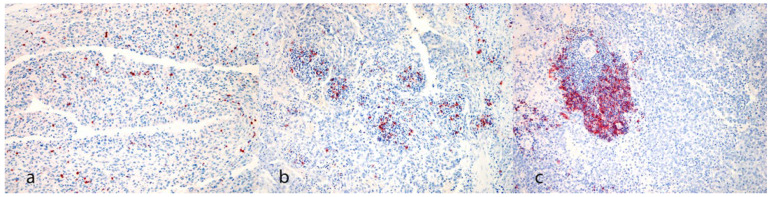
Immunohistochemical evaluation. (**a**) CD3 reaction pattern characterized by single positive cells scattered within neoplastic cells (case N. 20, 20×). (**b**) CD20 reaction pattern characterized by small aggregates of positive cells scattered within neoplastic cells (case N. 6, 20×). (**c**) CD20 reaction pattern characterized by massive aggregates (>30 + cells) among neoplastic cells (case N. 31, 20×) [(**a**–**c**) 3-Amino-9-Ethylcarbazole (AEC) chromogen; Carazzi’s hematoxylin counterstain].

**Figure 3 animals-14-01224-f003:**
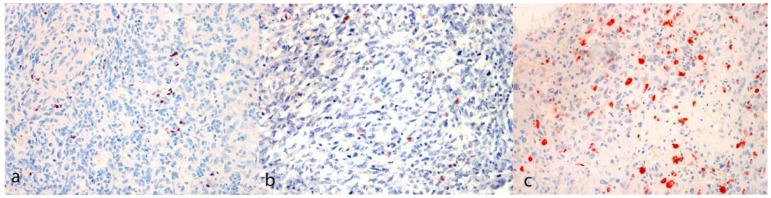
Immunohistochemical evaluation. (**a**) FoxP3 reaction pattern characterized by scattered positive cells among neoplastic cells (case N. 13, 40×). (**b**) CTLA-4 reaction pattern characterized by isolated positive neoplastic cells (case N. 22, 40×). (**c**) Iba-1 reaction pattern characterized by scattered positive cells among neoplastic cells (case N. 8, 40×) [(**a**–**c**) 3-Amino-9-Ethylcarbazole (AEC) chromogen; Carazzi’s hematoxylin counterstain].

**Figure 4 animals-14-01224-f004:**
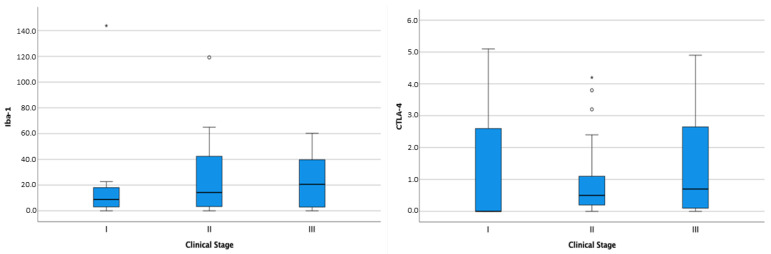
Box-and-whisker plots of Iba-1 and CTLA-4 distribution according to clinical stage. For each box, the central horizontal line represents the median, and the lower and upper boundaries represent the 25th and 75th percentiles, respectively. Whiskers represent the most extreme observations that were not outliers. Empty dots (◦) and asterisks (*) represent moderate and extreme outliers (i.e., values that are at a distance from the 25th or 75th percentile greater than 1.5 or 3 times the interquartile range, respectively).

**Figure 5 animals-14-01224-f005:**
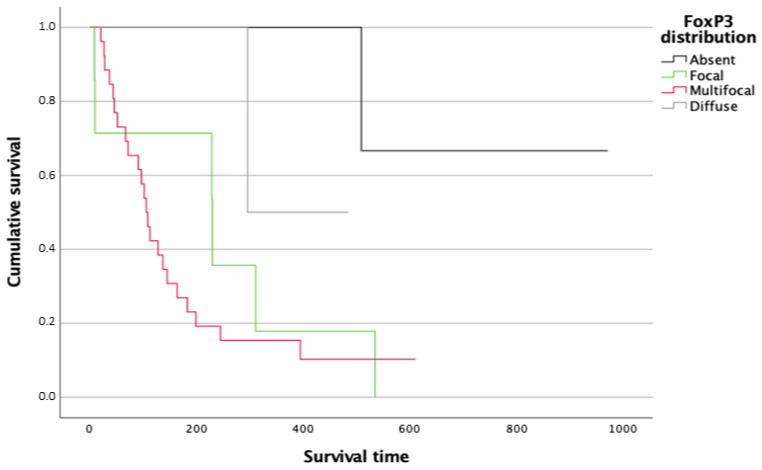
Kaplan–Meier survival curves for follow-up overall survival after initial histological diagnosis for dogs with splenic HSA according to the type of FoxP3+ cell distribution patterns.

**Figure 6 animals-14-01224-f006:**
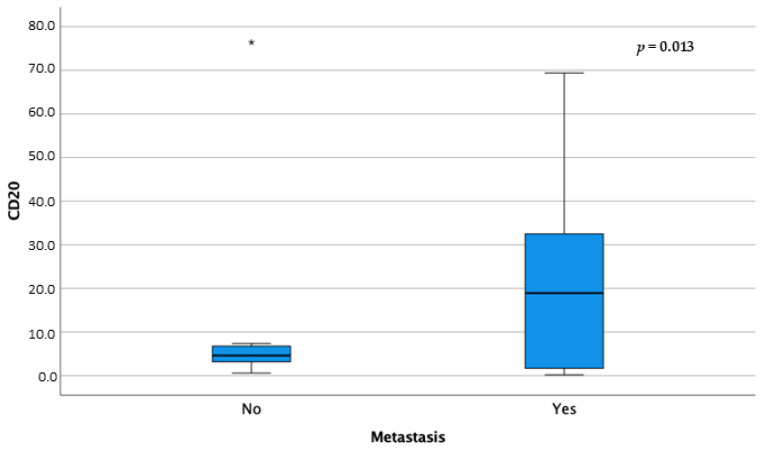
Box-and-whisker plots of CD20 distribution according to metastatic disease. For each box, the central horizontal line represents the median, and the lower and upper boundaries represent the 25th and 75th percentiles, respectively. Whiskers represent the most extreme observations that were not outliers. Asterisks (*) represent moderate and extreme outliers (i.e., values that are at a distance from the 25th or 75th percentile greater than 1.5 or 3 times the interquartile range, respectively).

**Figure 7 animals-14-01224-f007:**
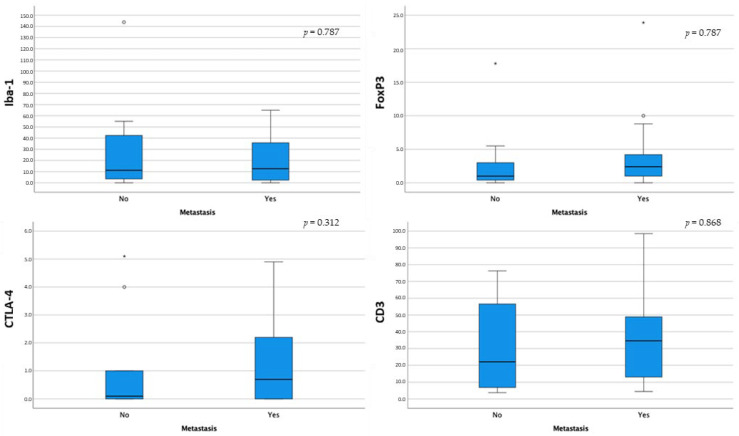
Box-and-whisker plots of Iba-1, FoxP3, CTLA-4, and CD3 distribution according to metastatic disease. For each box, the central horizontal line represents the median, and the lower and upper boundaries represent the 25th and 75th percentiles, respectively. Whiskers represent the most extreme observations that were not outliers. Empty dots (◦) and asterisks (*) represent moderate and extreme outliers (i.e., values that are at a distance from the 25th or 75th percentile greater than 1.5 or 3 times the interquartile range, respectively).

**Table 1 animals-14-01224-t001:** List of antibodies, suppliers, antigen retrieval methods, and dilutions.

Antibody	Type	Clone	Manufacturer	Antigen Retrieval	Dilution
Iba-1	Mouse monoclonal	MABN92	Merck Millipore (Burlington, MA USA)	HIER; Tris-EDTA; pH 9.0	1:100
FoxP3	Rat monoclonal	FJK-16s	eBioscience™ (# 14-5773-82) (San Diego, CA, USA)	HIER; Tris-EDTA; pH 9.0	1:100
CTLA-4	Mouse monoclonal	F-8	Santa Cruz Biotechnology (Dallas, TX, USA)	HIER; Tris-EDTA; pH 9.0	1:100
CD3	Rabbit polyclonal	-	Dako (A0452) (Santa Clara, CA, USA)	HIER; Tris-EDTA; pH 9.0	1:200
CD20	Rabbit polyclonal	-	Thermo Scientific (RB-9013) (Waltham, MA, USA)	No AR	1:200

## Data Availability

All study data are presented in the article.

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
