# Peer review of "Tumor Immune Microenvironment and Its Clinicopathological and Prognostic Associations in Canine Splenic Hemangiosarcoma"

_animals, 2024, doi:10.3390/ani14081224_

Round 1
Reviewer 1 Report (New Reviewer)
Comments and Suggestions for Authors
The manuscript is an interesting study about the clinicopathological and prognostic association of the microenvironment of canine splenic hemangiosarcoma. It is written clearly, describes the methodology, results, and discussion in detail. There are some suggestions to improve the manuscript:
Tables 2 and 3: both tables are very long, is it possible to simplify concentrating the data in a 1 or 2 graphics? For example or use the table as complementary data.
Results, lines 257 to 280, this information contrary to the previous data is better to present in a table and concentrate the most important results in the text.
Figure 2, there are many immunostainings in a one figure, maybe is possible to separate in 2 or 3 figures with one immunostaining or case in each one, with better definition.
Lines 484 to 490, this paragraph can be changed to the final conclusions, and delete the sentence in the line 486 (that in my opinion) it not a limitation for your study.
Line 492: this sentence is not very explicit, how the immune response can influence the canine 492 splenic HAS?
Author Response
We are truly grateful, for the evaluation given by the reviewer. Their valuable feedback and suggestions have improved the quality and clarity of our document. Following are the response to reviewer’s comments:
The manuscript is an interesting study about the clinicopathological and prognostic association of the microenvironment of canine splenic hemangiosarcoma. It is written clearly, describes the methodology, results, and discussion in detail. There are some suggestions to improve the manuscript:
Tables 2 and 3: both tables are very long, is it possible to simplify concentrating the data in a 1 or 2 graphics? For example or use the table as complementary data.
Tables 2 and 3 have been moved to supplementary tables and have been renumbered as Supplementary Table 1 (former Table 2) and Supplementary Table 2 (former Table 3).
Results, lines 257 to 280, this information contrary to the previous data is better to present in a table and concentrate the most important results in the text.
We thank this reviewer for this comment. However, we believe that the redundant part of the figure is relative only to the distribution (absent, focal, multifocal, diffuse) of the immune cells and not to the other features of them (number of cells/mm2; aggregation pattern; correlation coefficient). Therefore, we would like to keep both the figure and text, but we have deleted from the text the redundant lines regarding the distribution of immune cells.
Figure 2, there are many immunostainings in a one figure, maybe is possible to separate in 2 or 3 figures with one immunostaining or case in each one, with better definition.
We have separated the original plate in two different figures, eliminating two pictures.
Lines 484 to 490, this paragraph can be changed to the final conclusions, and delete the sentence in the line 486 (that in my opinion) it not a limitation for your study.
The indicated line has been removed and the take home message of this paragraph has been added to the final conclusion.
Line 492: this sentence is not very explicit, how the immune response can influence the canine 492 splenic HAS?
We modified the sentence and hopefully made it more clear. The meaning of the sentence was the demonstration of the presence of an immune response in canine HSA through the demonstration of the presence of different immune cell populations within the tumors.

Reviewer 2 Report (New Reviewer)
Comments and Suggestions for Authors
The article is interesting and raises a growing area of interest in oncology.
I think it is essential that the authors clarify from the summary that this is a retrospective study.
The authors must use the nomenclature related to cancer. The term "tumor features" is unclear to me; I suppose it relates to pathological grading. They must also clarify the relevance of clinical graduation since they incorporate it into the methodology without addressing the topic previously. Is there a theoretical framework related to it?
On line 22, they mention tumor-related death and survival. What difference do these variables have?
They should try to maintain the units of measurement (line 50, days, and months are used in the same variable)
In lines 54 to 55, they mention a grading system that is completely pathological; no mention is made of anatomical-clinical grading.
On lines 207 and 208 they mention experienced pathologists, are they commission certified, do they have an academic credential like a PhD? It is not mentioned in the authorship.
Tables 2 and 3 are unnecessary. The demographic and clinicopathological information can be represented in figures or analyzed statistically to understand the result. Something similar happens with Table 3; the results are seen and processed later.
Figure 1 and the paragraphs between lines 257 and 283 are repetitive.
Figure 2 requires improvement. The letters are not very visible; the magnification or distance bar should be in the figure. Ideally, all figures should have the same magnification or be ordered accordingly. In the figure caption, the chromatophore is not declared; it is assumed what the immunostaining is, but I think it is relevant that it be declared. Cell populations are not demarcated either.
Figure 3 does not present significant differences, so it can be eliminated and the result summarized in text. The same consideration should be taken with Figure 6.
I think they should further elaborate on the concept raised in line 379 in
the discussion about the role in the oncological progression of Tregs from the point of view of the mechanism and not just the association.
Author Response
We are truly grateful, for the evaluation given by the reviewer. Their valuable feedback and suggestions have improved the quality and clarity of our document. Following are the response to reviewer’s comments:
I think it is essential that the authors clarify from the summary that this is a retrospective study.
The term retrospective has been added in the simple summary, as well as in the abstract
The authors must use the nomenclature related to cancer. The term "tumor features" is unclear to me; I suppose it relates to pathological grading. They must also clarify the relevance of clinical graduation since they incorporate it into the methodology without addressing the topic previously. Is there a theoretical framework related to it?
The term “tumor features” has been specified as tumor’s histologic characteristics (line 21, line 36). Regarding the histologic grading of canine splenic HSA, this has been cited in the introduction because there was an attempt, by 2 studies, to apply a grading system. However, subsequent investigations did not corroborate the findings of these two studies. Consequently, even in subsequent papers, including the review article on grading systems in veterinary medicine (which were included as a reference in the paper as refs #17 and 18) it is noted that the histologic grading system for canine splenic HSA lacks prognostic significance and has therefore not been widely adopted. We did not adapt any grading system in our study either.
On line 22, they mention tumor-related death and survival. What difference do these variables have?
Regarding prognostic factors, tumor-related death and overall survival are related but distinct concepts. Tumor-related death refers to death occurring as a direct consequence of the tumor progression or its complications, whereas overall survival is the time from diagnosis (or start of treatment) until death, whatever the cause of the death is (including both tumor-related and tumor-unrelated causes). Overall survival gives an indication on how long patients with tumor live, independent on the specific cause of death.
They should try to maintain the units of measurement (line 50, days, and months are used in the same variable)
A consistent unit of measurement was adopted
In lines 54 to 55, they mention a grading system that is completely pathological; no mention is made of anatomical-clinical grading.
A comment to the clinical staging has been added in the introduction (lines 59-61).
On lines 207 and 208 they mention experienced pathologists, are they commission certified, do they have an academic credential like a PhD? It is not mentioned in the authorship.
Both authors are DVM, PhD and have 15 (IP) and 25 (MS) years’ experience in diagnostic pathology. IP is also ECVP boarded since 2019.
Tables 2 and 3 are unnecessary. The demographic and clinicopathological information can be represented in figures or analyzed statistically to understand the result. Something similar happens with Table 3; the results are seen and processed later.
As per reviewer’s 1 suggestions, tables 2 and 3 have been moved as supplementary tables.
Figure 1 and the paragraphs between lines 257 and 283 are repetitive.
We thank this reviewer for this comment. However, we believe that the redundant part of the figure is relative only to the distribution (absent, focal, multifocal, diffuse) of the immune cells and not to the other features of them (number of cells/mm2; aggregation pattern; correlation coefficient). Therefore, we would like to keep both the figure and text, but we have deleted from the text the redundant lines regarding the distribution of immune cells.
Figure 2 requires improvement. The letters are not very visible; the magnification or distance bar should be in the figure. Ideally, all figures should have the same magnification or be ordered accordingly. In the figure caption, the chromatophore is not declared; it is assumed what the immunostaining is, but I think it is relevant that it be declared. Cell populations are not demarcated either.
The letters have been magnified and the original plate has been divided into two separate plates, each containing three pictures of the same magnification. The magnification has been specified in the legend and we opted not to insert a bar in the images to prevent overcrowding, especially since there is also a letter present. Following the author’s instruction regarding image presentation, the inclusion of a magnification bar in the picture is not mandatory. Additionally, we provided details about the chromogen used in the legend. Regarding the suggestion to demarcate the positive cells, we would like to clarify that in the context of immunohistochemical images, cells are typically considered either positive (stained) or negative (unstained). As such, explicitly marking the positive cells with arrows is usually redundant and not commonly adopted in the scientific literature.
Figure 3 does not present significant differences, so it can be eliminated and the result summarized in text. The same consideration should be taken with Figure 6.
We appreciate the reviewer’s feedback and acknowledge that the results do not show statistically significant differences, as discussed in the text. However, we believe that the graphical representation provides a clearer illustration on the elevated median values among the clinical stages and the metastatic groups, respectively, for the immune cell categories. Given that the other two reviewers did not raise concerns about removing these figures, we propose retaining them in the paper.
I think they should further elaborate on the concept raised in line 379 in the discussion about the role in the oncological progression of Tregs from the point of view of the mechanism and not just the association.
The role of Tregs has been expanded in lines 378-382.
Reviewer 3 Report (New Reviewer)
Comments and Suggestions for Authors
Line 46: accumulating data now suggest a pluripotent bone marrow progenitor as the cell of origin for this disease. https://www.mdpi.com/2306-7381/2/4/388: authors may also wish to include this article in the references.
Line 54 and 55: The two references are not exactly a proposal grading system for canine splenic HSA. Most appropriate references should be chosen.
Line 179: They don't explain why they chose IBA-1 antibody to mark macrophages.
Figure 2: Lowercase letters in images and capitals in captions: must be standardised.
Author Response
We are truly grateful, for the evaluation given by the reviewer. Their valuable feedback and suggestions have improved the quality and clarity of our document. Following are the response to reviewer’s comments:
Line 46: accumulating data now suggest a pluripotent bone marrow progenitor as the cell of origin for this disease. https://www.mdpi.com/2306-7381/2/4/388: authors may also wish to include this article in the references.
The reference has been added to the sentence and the sentence has been expanded accordingly.
Line 54 and 55: The two references are not exactly a proposal grading system for canine splenic HSA. Most appropriate references should be chosen.
All cited papers refer to the original attempts to apply a histologic scheme to predict the behavior of HSA in dogs, notably by Moore and Ogilvie. Moore references Ogilvie’s paper in the introduction, discussing the initial report of histologic grading as prognostic factor. While the authors of this study are not aware of any other paper focusing specifically on the grading system, we are willing to include any relevant references that the reviewer would like to suggest. However, considering that the main aim of the papers by Moore and Ogilvie was not to establish a grading scheme, but rather to assess the prognostic potential of some histologic characteristics, the sentence has been rephrased accordingly.
Line 179: They don't explain why they chose IBA-1 antibody to mark macrophages.
The reason why we chose Iba-1 antibody as a pan-macrophage marker is detailed in lines 471-480.
Figure 2: Lowercase letters in images and capitals in captions: must be standardised.
Letters have been standardized in the captions.
This manuscript is a resubmission of an earlier submission. The following is a list of the peer review reports and author responses from that submission.
Round 1
Reviewer 1 Report
Comments and Suggestions for Authors
In the article "tumor immune microenvironment and its clinicopathological and prognostic associations in canine splenic hemangiosarcoma" the authors perform a retrospective observational study in which they analyze the presence and distribution of tumor-infiltrating lymphocytes (TILs) in 56 cases of canine splenic hemangiosarcoma.
Both the introduction and discussion are very well documented and they even make comparisons between canine and human tumors. The immunohistochemistry has been correctly performed as the images are clear, although I am left in doubt as to whether the authors have quantified the number of cells at the magnification shown in Figure 2, as they have no scale nor are the magnifications used indicated in the figure caption.
I liked the fact that the authors themselves acknowledge the limitations of their study in line 447, but it is precisely these limitations that make the article not contribute anything new to what is already known. The authors' final conclusion is that a higher presence of Treg is associated with a worse prognosis and this has been demonstrated for some time in different types of tumors and species. Furthermore, on line 374 the authors say "the tumor burden, PD-L1 overexpression, and an immnunogenic tumor microenvironment have been suggested to predict the response to immunotherapy, no study demonstrated the efficacy of any single parameter alone, suggesting that an integrated approach using multiple targets in combination should be used". I very much agree with that statement, but I do not understand why the authors then do this study in which only one cell type of the tumor microenvironment is studied in depth. The microenvironment is a very complex structure in which there are many cell types that interact with each other in which small differences favor or inhibit tumor growth.
In my opinion, although the theoretical framework is well founded, the authors should make a much more complete study of the microenvironment if they want to publish the article. Otherwise, it does not contribute anything novel.
Reviewer 2 Report
Comments and Suggestions for Authors
In this manuscript, the authors revealed the role of the microenvironment in the process of HSA and highlighted the relationship between tumor-infiltrating lymphocytes with events related to survival and metastasis. My comments are as follows:
1. More details related to FoxP3 and CD20 are suggested to be added in the introduction as the background. And the second paragraph of the introduction is complicated to read. Separated from line 67 would be a better structure.
2. Is that possible to provide a multiple-factor analysis of variance to reveal the importance of FoxP3 or CD20?
3. In Figure 3, the quantity of CTLA-4 looked like decreased with the clinical stage.
4. The author did not provide clear data to support the effect of CD20+ cells to survival.